# The Pleiotropic Role of Vitamin K in Multimorbidity of Chronic Obstructive Pulmonary Disease

**DOI:** 10.3390/jcm12041261

**Published:** 2023-02-05

**Authors:** Ianthe Piscaer, Rob Janssen, Frits M. E. Franssen, Leon J. Schurgers, Emiel F. M. Wouters

**Affiliations:** 1Department of Respiratory Medicine, Maastricht University Medical Centre (MUMC+), 6200 MD Maastricht, The Netherlands; 2Department of Respiratory Medicine, Canisius-Wilhelmina Hospital, 6532 SZ Nijmegen, The Netherlands; 3Department of Research and Development, CIRO+, Centre of Expertise for Chronic Organ Failure, 6085 NM Horn, The Netherlands; 4Department of Biochemistry, Cardiovascular Research Institute Maastricht, 6200 MD Maastricht, The Netherlands; 5Ludwig Boltzmann Institute for Lung Health, 1140 Vienna, Austria

**Keywords:** chronic obstructive pulmonary disease, emphysema, vitamin K, matrix Gla-protein, cardiovascular disease, chronic kidney disease, osteoporosis, sarcopenia

## Abstract

Although defined by the presence of airflow obstruction and respiratory symptoms, patients with chronic obstructive pulmonary disease (COPD) are characterized by multimorbidity. Numerous co-occurring conditions and systemic manifestations contribute to the clinical presentation and progression of COPD; however, underlying mechanisms for multimorbidity are currently not fully elucidated. Vitamin A and vitamin D have been related to COPD pathogenesis. Another fat-soluble vitamin, vitamin K, has been put forward to exert protective roles in COPD. Vitamin K is an unequivocal cofactor for the carboxylation of coagulation factors, but also for extra-hepatic proteins including the soft tissue calcification inhibitor matrix Gla-protein and the bone protein osteocalcin. Additionally, vitamin K has been shown to have anti-oxidant and anti-ferroptosis properties. In this review, we discuss the potential role of vitamin K in the systemic manifestations of COPD. We will elaborate on the effect of vitamin K on prevalent co-occurring chronic conditions in COPD including cardiovascular disorders, chronic kidney disease, osteoporosis, and sarcopenia. Finally, we link these conditions to COPD with vitamin K as a connecting factor and provide recommendations for future clinical studies.

## 1. Introduction

Chronic obstructive pulmonary disease (COPD) is characterized by chronic respiratory symptoms and persistent airflow limitation [1]. Structural abnormalities in the lungs are part of the disease, including parenchymal destruction and small airways disease, and one of both may predominate relative to the other [1]. In addition, patients with COPD frequently have multiple co-occurring chronic conditions (i.e., multimorbidity) [2]. Many of these are more common and show more overlap in COPD patients compared with controls [3]. Recognition and treatment of these disorders are relevant since they have a negative impact on the quality of life, morbidity, hospitalizations, and mortality in COPD patients [4].

Smoking is the most important risk factor for the development of COPD and is causally related to multiple chronic morbidities including cardiovascular disease [1,4]. Smoking is, however, not solely accountable for the high prevalence of other chronic diseases in COPD patients. Several additional underlying mechanisms have been suggested, including systemic inflammation, oxidative stress, and genetic susceptibility [4,5,6,7]. Recently, the possible mechanistic role of vitamins in COPD has gained increasing attention. Especially fat-soluble vitamins A [8] and vitamin D [9] have been reported to exert positive effects on COPD. Vitamin K is another fat-soluble vitamin and is proposed to have favorable effects on bone and cardiovascular health [10]. In this review, we discuss the role of vitamin K in the multimorbidity of COPD (Figure 1).

## 2. Vitamin K

Vitamin K is well-known for its role as a cofactor in the carboxylation of coagulation factors II, VII, IX, and X as well as proteins C and S in the liver [10,11]. Furthermore, numerous extra-hepatic proteins are vitamin K-dependent [11]. Vitamin K is required for γ-carboxylation of glutamate residues of vitamin K-dependent proteins (VKDPs) thereby gaining biological activity, during a process that is catalyzed by the enzyme γ-glutamyl carboxylase (GGCX) [11]. As part of this conversion, vitamin K hydroquinone is oxidized into vitamin K epoxide [12]. The enzyme vitamin K epoxide reductase (VKOR) plays a critical role in the recycling of vitamin K by converting vitamin K epoxide back to reduced vitamin K in a process called the vitamin K cycle (Figure 2) [11].

Vitamin K1 (phylloquinone) and vitamin K2 (menaquinones) are the two naturally occurring forms of vitamin K [11]. Green leafy vegetables are an important source of vitamin K1, whereas vitamin K2 is produced by bacteria and mainly found in fermented food products including cheese and the Japanese traditional breakfast “natto” consisting of soybeans fermented by Bacillus subtilis [11,13]. Vitamin K1 is preferentially taken up in the liver, where it is used to activate coagulation factors [11]. Long-chain vitamin K2 is more lipophilic and, due to the long plasma half-life, more efficaciously penetrates extra-hepatic tissues for activation of matrix Gla-protein (MGP) in blood vessels and osteocalcin in bone [11,14]. In the Western diet, the majority of vitamin K intake occurs in the form of vitamin K1 [15]. Vitamin K2, however, has a higher bioavailability than vitamin K1 and may therefore be at least as important for its bioactivity [15,16].

Vitamin K status is influenced by different factors (Figure 3). Dietary intake is the most important of these; however, vitamin K is also synthesized by intestinal bacteria [10]. Poor intake, malabsorption, and reduction of intestinal bacteria by the use of antibiotics are factors that have a negative influence on vitamin K status [10]. In addition, vitamin K status is affected by the efficacy of vitamin K recycling. Various single nucleotide polymorphisms (SNPs) in the VKOR complex subunit 1 (VKORC1) exist, with some haplotypes resulting in lower vitamin K recycling than others [17]. The use of vitamin K antagonists (VKA) interferes with the recycling of vitamin K since these anticoagulants bind to and competitively inhibit the binding of vitamin K to VKORC1 [18]. Finally, vitamin K status can be reduced by increased utilization of vitamin K.

The daily recommended vitamin K intake varies per country and is usually set between 50 μg and 120 μg per day [10,19]. There are currently no cut-off values defining vitamin K deficiency. Although it is technically possible to measure vitamin K directly in circulation, several factors limit the use of serum concentrations as a biomarker for vitamin K status. Vitamin K1 levels in serum highly fluctuate and depend on vitamin K intake as well as triglyceride levels [19]. Vitamin K2 can usually not be detected in the blood when subjects are on an average Western diet unless vitamin K2-containing supplements are used [19]. Vitamin K deficiency in humans results in the secretion of biologically inactive, undercarboxylated forms of VKDPs into plasma [12]. Plasma levels of these proteins are commonly used for assessing *functional* vitamin K status, with high levels of undercarboxylated VKDPs reflecting low vitamin K status and vice versa. Undercarboxylated coagulation factor II is referred to as protein induced by vitamin K antagonism or absence II (PIVKA-II) and reflects hepatic vitamin K status. The inactive form of the calcification inhibitor MGP, i.e., dephosphorylated undercarboxylated (dp-uc) MGP, is widely used as a biomarker for extra-hepatic vitamin K status [12]. Undercarboxylated osteocalcin can be used for the same purpose [19]. On average, 20% to 30% of MGP and osteocalcin circulate in the undercarboxylated form in subjects not supplemented with vitamin K [16]. This indicates that a certain degree of subclinical vitamin K deficiency is highly prevalent. The triage theory postulates that in a state of shortage, vitamins are primarily directed to the most vital pathways in the body [20]. This implies that when supplies are low, vitamin K is preferentially used for the activation of coagulation factors in the liver at the expense of the activation of extra-hepatic VKDPs [20]. Since only carboxylated VKDPs harbor full activity, suboptimal vitamin K status potentially contributes to pathologies in different organ systems.

## 3. Vitamin K in COPD

Studies on the role of vitamin K in COPD patients are limited. A cross-sectional study using data from 17,681 subjects from the National Health and Nutrition Examination Survey (NHANES) demonstrated an inverse association between the presence of emphysema and vitamin K consumption [21]. The likelihood of having emphysema was 39% lower in subjects who met the recommended daily intake for vitamin K compared to subjects who did not, and significance was maintained after adjustment for confounders such as smoking, age, and body mass index (BMI) [21].

Another study demonstrated that dp-ucMGP levels (inversely corresponding with vitamin K status) were significantly higher in COPD patients compared to both smoking and never-smoking controls [22]. Dp-ucMGP was inversely associated with the diffusing capacity of the lung for carbon monoxide (DLCO) [22], suggesting an association between vitamin K status and emphysema. However, this correlation was not confirmed in a second independent cohort of COPD patients [22]. Additional research, including studies on the correlation between vitamin K status and emphysema quantification on computed tomography (CT), is necessary to further explore this association. Mortality was higher in COPD patients in the quartile with the highest dp-ucMGP levels compared to patients in the other three quartiles [22]. In line, preliminary retrospective data demonstrated that mortality was higher in COPD patients using VKA compared to those using direct oral anticoagulants (DOAC) [23]. It is tempting to speculate that VKA-induced vitamin K deficiency and, consequentially, insufficient extra-hepatic VKDP carboxylation might be the underlying mechanism contributing to higher mortality. It should be kept in mind that—although analyses were adjusted for possible confounders such as age and comorbidities—confounding by indication might be a source of bias in retrospective studies.

Although aforementioned studies suggest that vitamin K has favorable effects in COPD, specifically in emphysema, a causative relationship cannot be unequivocally proven based on the present limited data. Studies on the effect of vitamin K supplementation in COPD patients are recommended to further explore the beneficial effects of vitamin K on the progression of emphysema and other outcomes in COPD.

## 4. Cardiovascular Diseases

COPD patients have an almost 2.5-fold higher risk of cardiovascular disease in comparison with subjects without COPD [24]. The severity of airflow limitation is associated with both cardiovascular events as well as cardiovascular-related mortality independent of shared risk factors [7,25]. Vascular and heart valve calcification are among the mechanisms that contribute to cardiovascular diseases. Calcification is an active process that is promoted by factors such as pro-inflammatory cytokines and counteracted by anti-calcifying proteins [26]. MGP is predominantly synthesized by vascular smooth muscle cells and is one of the most potent inhibitors of vascular calcification [26,27]. To execute anti-calcifying properties, the Gla-residues of activated MGP have a strong affinity for calcium crystals [28]. Besides γ-carboxylation, MGP can undergo serine-phosphorylation as an additional post-translational modification [28]. The role of phosphorylation is currently not fully clarified; however, it has been shown that phosphorylation also enhances the binding of MGP to calcium crystals by generating negatively charged phosphoserine residues [29,30]. The importance of MGP was demonstrated in mice lacking MGP, which all died in the first weeks of life as a consequence of large blood vessel ruptures due to extensive vascular mineralization [31]. Two types of vascular calcification can be distinguished, i.e., medial and intimal calcification. Medial arterial calcification usually starts with calcium deposition along the elastic fibers of the medial layer and ultimately results in arterial stiffness and left ventricular hypertrophy [12,32]. Patients with chronic kidney disease (CKD) and diabetes mellitus are particularly prone to medial arterial calcification, and this process also occurs as part of aging [12]. VKA-induced vitamin K deficiency resulted in extensive medial elastocalcinosis in a rat model and, importantly, this process could be reversed by a vitamin K-rich diet [33]. In human arteries with medial calcification, undercarboxylated MGP was mainly found at sites of calcification whereas MGP was solely found in the carboxylated (i.e., active) form in the tunica media of healthy vascular tissue [34]. Intimal arterial calcifications are associated with atherosclerotic plaques [35]. Treatment with VKA increased both atherosclerotic plaque calcification and plaque progression in mouse models [36,37]. The expression of carboxylated MGP in these plaques was decreased, whereas there was an increase in undercarboxylated MGP in calcified areas [36,37]. In line, in the intimal layer of human atherosclerotic arteries, undercarboxylated MGP was mainly detected at sites of microcalcification [34,38]. These studies suggest that vitamin K deficiency is a risk factor for developing medial as well as intimal arterial calcification through compromised local MGP carboxylation.

Human observational studies demonstrated that lower consumption of vitamin K2, but not vitamin K1, was associated with increased coronary calcification [39], severe aortic calcification [40], increased risk of coronary heart disease (CHD) [41], and CHD-related mortality [40]. High dp-ucMGP levels were associated with increased pulse wave velocity (PWV) [42,43], a surrogate marker of cardiovascular risk, and with increased cardiovascular mortality in subjects with stable vascular disease [44] and diabetes mellitus type 2 [45]. A meta-analysis showed that high dp-ucMGP levels were associated with an increase in combined endpoint ‘cardiovascular disease and mortality’; however, the correlation lost significance when each of the endpoints were analyzed separately [46].

Several studies investigated the effects of vitamin K supplementation on cardiovascular outcomes. Studies in patients with CKD will be discussed in the following paragraph. In postmenopausal women, three years of supplementation with vitamin K1 significantly enhanced the compliance and distensibility of the arterial vessel wall [47]. There was no effect, however, on the progression of atherosclerosis quantified by carotid intima-media thickness (cIMT) [47]. In healthy postmenopausal women, three years of supplementation of vitamin K2 was effective in reducing arterial stiffness quantified by PWV [48]. In addition, elastic properties of the carotid artery improved in the subgroup of women with high arterial stiffness at baseline [48]. In contrast, there was no effect in a trial of 6-month vitamin K2 supplementation on PWV, endothelial function quantified by flow-mediated dilatation (FMD) of the brachial artery and cIMT [49]. A meta-analysis combining these three studies concluded that there was no effect of vitamin K supplementation on arterial stiffness [46]. It could be argued, however, whether the last-mentioned trial was long enough in duration to reveal positive effects on arterial stiffness. In line, the study of Knapen et al. showed that a positive effect on arterial stiffness was not found before the third year of supplementation [48].

Vitamin K1 supplementation attenuated the progression of coronary artery calcium (CAC) in elderly adults in three years, but only in the subgroup adherent to therapy [50]. Vitamin K1 did not prevent the development of new CAC lesions [50]. Another trial demonstrated significant deceleration of aortic valve calcification progression after 12 months of vitamin K1 supplementation [51]. In contrast, 2-year vitamin K2 and vitamin D supplementation in subjects with aortic stenosis did not slow the progression of aortic valve calcification [52]. Preliminary results of a sub-analysis of this study showed that in subjects with high CAC at baseline and no history of coronary disease, combined vitamin K2 and vitamin D supplementation significantly slowed down CAC progression [53]. In contrast, two trials with a shorter 6-month intervention of vitamin K2 supplementation in subjects with diabetes mellitus did not find a significant effect on arterial calcification [54,55].

Of note, considerable methodological diversity between trials exists regarding factors such as the co-administration of other vitamins, duration of supplementation, patient characteristics including comorbidities and age, study population size, concealing of treatment allocation as well as vitamin K dosage and form. These factors could at least be partly responsible for some contradicting results.

In conclusion, most observational studies demonstrate that reduced vitamin K status is associated with increased arterial stiffness and vascular calcification as well as with a higher risk of fatal and non-fatal cardiovascular events. Taking the clinical trials together, there might be a protective effect of long-term vitamin K supplementation on arterial stiffness and both arterial and heart valve calcification, particularly in patients with pre-existing arterial stiffness and calcification. Both pathological conditions are highly prevalent in COPD [56,57] and vitamin K status could potentially be a mechanistic link and a promising therapeutic target. The underlying pathobiological mechanisms, however, still have to be determined. Further, data from experimental animal studies suggest harmful effects of vitamin K deficiency on atherosclerotic plaque formation through insufficient MGP carboxylation [36,37]. Yet, a direct effect of vitamin K supplementation on atherosclerosis in human clinical trials was not demonstrated. None of the supplementation trials included clinical outcome parameters such as the incidence of cardiovascular events or cardiovascular mortality.

## 5. Chronic Kidney Disease

CKD is defined as a glomerular filtration rate (GFR) below 60 mL/min/1.73 m^2^ and/or the presence of kidney damage (e.g., albuminuria), persisting for more than 3 months [58,59]. CKD is a manifestation of multimorbidity in COPD that is frequently underrecognized. However, CKD occurs 2.2 times more frequently in COPD patients compared to subjects without COPD [60] and the prevalence of reduced kidney function defined by an estimated GFR (eGFR) less than 60 mL/min/1.73 m^2^ exceeds 20% in the COPD population [61].

CKD patients are at increased risk of developing vitamin K deficiency. This is in part the consequence of the low-potassium and low-phosphorus diet that is recommended to these patients. This diet is usually low in vitamin K through avoidance of green leafy vegetables and dairy products, sources rich in vitamin K1 and K2, respectively [62]. In addition, vitamin K recycling might be hampered in a uremic environment and drugs such as phosphate binders may have a negative impact on vitamin K status [63,64]. CKD is frequently accompanied by the presence of vascular calcification [12] and, in order to inhibit this process, there is a greater demand for vitamin K to activate MGP, thereby further depleting vitamin K status [62].

It has been consistently reported that dp-ucMGP levels inversely associate with eGFR [65,66] and that dp-ucMGP levels increase with more advanced CKD stages [29,67,68]. In a longitudinal study in the general population, high plasma dp-ucMGP levels were associated with an increased risk of developing kidney dysfunction and microalbuminuria [69]. Another prospective study confirmed this; however, significance was lost after correction for baseline kidney function and age [66]. It has been proposed that dp-ucMGP levels are high in CKD patients as a consequence of impaired glomerular filtration of dp-ucMGP rather than as a reflection of vitamin K deficiency [70]. Arguments against this hypothesis are, however, the observation that renal fractional MGP extraction was independent of kidney function in hypertensive patients [71] and the finding that total undercarboxylated MGP (i.e., the total of phosphorylated and dephosphorylated undercarboxylated MGP) was positively associated with eGFR [72]. It has been demonstrated that VKA can induce nephropathy [73], although it is currently not clear whether this effect is vitamin K-related or a consequence of over-anticoagulation irrespective of the type of anticoagulant. Mechanisms by which vitamin K could potentially attenuate kidney function deterioration are MGP-dependent protection of soft tissue calcification in the kidneys and prevention of renal interstitial fibrosis [69].

CKD patients are prone to vascular calcification due to disturbed mineral regulation [12]. The presence of vascular calcifications in CKD patients is associated with six times the increase in risk for cardiovascular events [74]. Vitamin K could be particularly of interest in CKD patients through optimizing the activation of MGP as an inhibitor of calcification. A prospective study in patients at different stages of CKD demonstrated that reduced total vitamin K consumption was associated with higher cardiovascular mortality [75]. In cross-sectional studies, dp-ucMGP levels were positively associated with the severity of vascular calcification [29,67] and PWV [68] in CKD patients. Not all studies, however, found associations between dp-ucMGP levels and cardiovascular endpoints in the CKD population [76].

In contrast to observational studies, vitamin K2 supplementation trials have shown disappointing results. One-year supplementation of vitamin K2 did not improve PWV in patients with CKD stage 3b or 4 (eGFR 15–45 mL/min/1.73 m^2^) [77]. In another trial, supplementation of vitamin K2 for 9 months in patients with non-dialysis requiring CKD stage 3 to 5 (eGFR < 60 mL/min/1.73 m^2^) resulted in an attenuation of cIMT progression; however, the effect on deceleration of CAC was only modest and non-significant [78]. Three trials of 12-, 18- and 24-month supplementation of vitamin K2, respectively, did not demonstrate a decline in progression of vascular calcification [79,80,81] and arterial stiffness [81] in hemodialysis patients. A possible explanation for the predominantly negative results could be that the uptake and transportation of vitamin K2, specifically in the form of menaquinone-7, is different in a uremic environment [63]. In kidney transplant recipients, uremic toxin levels usually decrease [82] and recent preliminary results of a randomized controlled trial (RCT) demonstrated that only 12 weeks of vitamin K2 supplementation resulted in a significant improvement of PWV in this patient population [83].

In contrast to vitamin K2, no altered pharmacokinetics in a uremic environment was shown for vitamin K1 [84]. Therefore, vitamin K1 may be more effective in CKD. Indeed, a recent trial in hemodialysis patients with pre-existing CAC demonstrated that high dose vitamin K1 supplementation for the duration of 18 months resulted in a significant deceleration of thoracic aortic calcium (TAC) progression and a trend towards an attenuation of CAC progression [85]. Other trials of vitamin K1 supplementation in the CKD population are currently ongoing [84].

Another explanation for predominantly negative trials in CKD might be that vascular calcification in kidney disease is multifactorial in origin and that solely enhancing vitamin K status -leaving concentrations of other trace elements such as phosphorus and calcium unaltered- is mostly insufficient to counteract the process [84]. In line, an experimental animal model of vitamin K-deficient rats with kidney failure demonstrated that treatment with the combination of vitamin K2 and phosphate binders was significantly more effective in reducing aortic calcification compared to one of both treatments alone [86].

In conclusion, subjects with CKD are at increased risk for vitamin K deficiency. Reduced vitamin K status is associated with poor kidney function, although it remains unclear if a causal relationship exists and whether vitamin K supplementation could avert kidney function loss. Impaired GFR and microalbuminuria are manifestations of microvascular disease involvement in COPD and vitamin K could be a mechanistic link between these disorders. Reduced vitamin K status is associated with unfavorable cardiovascular effects in CKD patients in observational studies. Given that clinical trials have failed so far to demonstrate beneficial effects, there is currently no evidence for vitamin K2 supplementation in CKD. To date, only one trial investigated the effect of vitamin K1 supplementation in CKD patients, demonstrating positive and promising effects on arterial calcification. Additional RCTs of vitamin K1 supplementation in CKD patients are currently ongoing and could give more insight.

## 6. Osteoporosis

Osteoporosis is characterized by a combination of decreased bone mass and alterations in bone quality, resulting in increased bone fragility and thereby higher susceptibility to fractures [87]. The World Health Organization (WHO) defines osteoporosis as bone mineral density (BMD), measured by Dual-energy X-ray absorptiometry (DXA) scanning, of more than 2.5 standard deviations below the mean for young adults [88]. The reported prevalence of osteoporosis in COPD varies from 36% to 60% and is particularly high in severe disease [89]. Osteoporosis-related bone fractures have an important impact on COPD contributing to deconditioning, morbidity, and mortality [89]. Vitamin K has received less attention than vitamin D for its role in bone health, but a deficiency of vitamin K may also be a potential link between COPD and osteoporosis. Osteoblasts synthesize vitamin K-dependent osteocalcin, which is the most plentiful non-collagenous protein in bone tissue [90,91]. After vitamin K-dependent carboxylation, osteocalcin contributes to both formation of hydroxyapatite crystals in bone and the inhibition of bone mineralization, thereby preventing over-mineralization [91]. MGP is also involved in both processes of bone mineral regulation [91]. In addition, vitamin K directly affects bone metabolism, e.g., by protecting against reactive oxygen species (ROS) that are produced by osteoblasts [90,91].

In a recent meta-analysis assessing the association between vitamin K1 intake and fracture risk, five studies were included with a total of 1114 fractures in 80,982 subjects [92]. Subjects with the highest vitamin K1 intake had a 22% lower fracture risk compared to subjects with the lowest intake; an increase in daily vitamin K1 intake of 50 μg resulted in a 3% decrease in fracture risk [92]. With respect to the association between vitamin K consumption and BMD, the results are conflicting. Some studies did not find an association [93,94,95], whereas another study reported a positive correlation in women [96]. Besides low vitamin K intake, vitamin K deficiency can also be induced by the use of VKA. In a systematic review, VKA use was, however, not associated with fracture risk [97]. High serum levels of undercarboxylated osteocalcin (ucOC)—reflecting low vitamin K status—were associated with increased hip fracture risk [98,99] and reduced BMD of the hip [100].

Several clinical trials assessed the effect of vitamin K supplementation on fracture risk. The largest study in 4378 postmenopausal women with osteoporosis found no effect of 36-month vitamin K2 supplementation on the prevention of fractures [101]. However, in a subgroup of high-risk women with at least five pre-existing vertebral fractures, a significant protective effect on new vertebral fractures was found [101]. A meta-analysis combining the results of nine trials reported that the number of clinical fractures was 28% lower in postmenopausal and osteoporotic subjects when vitamin K was supplemented; however, significance was lost when restricting analyses to the five studies assessed as having a low risk of bias [102]. There were insufficient studies to draw definite conclusions in other subgroups and no effect was demonstrated on vertebral and hip fracture risk [102]. After two years of supplementation, there was a significant but not clinically relevant increase in BMD [102]. Another recent meta-analysis of vitamin K2 supplementation in postmenopausal women showed no effect on clinical fractures [103]; however, after excluding one heterogeneous study, a significant protective effect was found. Although this finding should be interpreted with caution given the risk of selection bias [103], it highlights that additional large and well-designed RCTs on vitamin K supplementation are warranted to evaluate the effect of vitamin K on the prevention of bone fractures.

It should be noted that vitamin K was administered in combination with other vitamins or minerals in some of the included studies. It has also to be mentioned that a high proportion of studies on the role of vitamin K in osteoporosis have been performed in Japanese populations. This may well be the reason for inconsistency with results from European-American populations [102].

In conclusion, data suggest a protective effect of vitamin K on fractures in high-risk subjects including subjects with pre-existing fractures, postmenopausal women, and subjects with osteoporosis. Additional RCTs are, however, warranted to confirm this. With regard to COPD, these findings are of particular interest since osteoporosis-related fractures considerably contribute to morbidity and mortality. Effects on BMD are inconsistent and it might therefore be that vitamin K exerts its protective effects on fractures by improving bone quality rather than BMD [104].

## 7. Sarcopenia

Sarcopenia is defined by low muscle strength, low muscle quantity or quality, and/or low physical performance and is associated with an increased likelihood of adverse outcomes including falls, fractures, physical disability, and mortality [105]. A recent meta-analysis reported that the prevalence of sarcopenia is as high as 22% in COPD patients [106]. Vitamin K might have a favorable role in sarcopenia by several proposed underlying mechanisms. Vitamin K has the capacity to stimulate vascular smooth muscle differentiation, has favorable effects on arteries thereby improving muscle perfusion [107], and functions as an electron carrier in skeletal muscle mitochondria [43], all of which might optimize muscle functioning. Vitamin K is also proposed to indirectly influence physical performance, due to its beneficial effects on inflammation and osteoarthritis [108].

Subjects with sarcopenia have a lower dietary intake of vitamin K compared to subjects without sarcopenia [109]. However, these results must be interpreted carefully, since they could indicate an overall reduced nutritional status in sarcopenic subjects rather than a vitamin K deficiency per se. Cross-sectional studies demonstrated that vitamin K status associates with markers for muscle strength. A correlation was found between high dp-ucMGP levels and a lower axial skeletal muscle mass in subjects with hypertension [43]. In a study of 1089 elderly subjects, high dp-ucMGP levels were associated with poorer lower extremity function reflected by the short physical performance battery (SPPB) and isokinetic leg strength, but not with walking endurance or decline in physical performance during 4–5 years of follow-up [108]. In 633 adults aged 55–65 years, high plasma dp-ucMGP was associated with decreased handgrip strength and smaller calf circumference and, in women, with a lower functional performance score [107]. Analogously to the previous study, no correlation was found between dp-ucMGP and change in functional parameters over time [107].

Three trials investigated the effect of vitamin K supplementation on parameters of physical function [49,110,111]. A 3-year RCT on vitamin K1 supplementation in elderly adults did not demonstrate an effect on muscle mass [110]. In another trial, 6-month supplementation of vitamin K2 had no effect on handgrip strength, or lower extremity function reflected by the SPPB and postural sway [49]. It should be mentioned that these were post hoc analyses, and that both trials were originally designed for other research questions. In a trial including 95 elderly subjects with a history of falls, supplementation with vitamin K2 for the duration of one year had no effect on anteroposterior sway or markers for physical function including the SPPB [111]. Of note, anteroposterior sway was already low at baseline in these subjects and it is arguable whether or not a further reduction of this parameter could be achieved [111]. It might be possible that vitamin K supplementation could have positive effects in subjects at higher risk or when other outcome measurements are selected.

In conclusion, vitamin K status is associated with markers for muscle function in cross-sectional, but not in longitudinal studies. The limited number of clinical trials that have been performed did not demonstrate favorable effects of vitamin K supplementation on physical performance. Therefore, it remains unknown whether the association between vitamin K and muscle function is causative in nature and whether the role of vitamin K in COPD-related sarcopenia has to be expected.

## 8. Discussion

Vitamin K status is reduced in COPD [22] and low vitamin K intake is associated with an increased risk of emphysema [21]. COPD is characterized by a high prevalence of co-occurring chronic conditions. Cardiovascular disease, CKD, osteoporosis, and sarcopenia are among the disorders that are highly prevalent in COPD and vitamin K may have a combined beneficial influence. The majority of vitamin K supplementation trials demonstrated positive effects on vascular calcification and arterial stiffness, especially in high-risk patients. In addition, vitamin K supplementation may be effective in reducing the risk of bone fractures. It should be noted that the effects of vitamin K on cardiovascular disease, CKD, osteoporosis, and sarcopenia have not been investigated in COPD patients per se. Although we would expect that the results from previous studies may be extrapolated to the COPD population, this of course has to be confirmed in future clinical studies specifically targeted to evaluate this subject.

The possible mechanisms through which vitamin K might exert positive effects in the lungs still remain elusive. Taking the available literature, the beneficial effects of vitamin K in the vasculature seem to be predominantly MGP-mediated, and we hypothesize that this might also account for its pulmonary effects. MGP is highly expressed in lungs [112], and pulmonary MGP expression can be upregulated by vitamin D [113]. COPD is characterized by accelerated elastic fiber degradation due to a disturbed protease and anti-protease balance [1]. Partially degraded elastic fibers are prone to calcification and calcified elastin is more rapidly degraded due to the upregulation of matrix metalloproteinases (MMPs) [114,115]. MGP inhibits elastic fiber calcification [12] and may therefore lower the vulnerability of elastic fibers to degradation. Indeed, high dp-ucMGP levels were associated with increased elastin degradation rate in four independent cohorts, including two cohorts of COPD patients [22,116,117]. Studies on lung tissue of COPD patients and healthy subjects, using conformation-specific MGP antibodies [34], could provide further insight into the localization of carboxylated and undercarboxylated MGP in the pulmonary system and their potential role in COPD pathogenesis. Moreover, vitamin K2 supplementation reduced serum MMP-3 levels in subjects with rheumatoid arthritis [118], which may also support the concept of vitamin K as an inhibitor of elastin degradation [119]. It has previously been postulated that elastin degradation is not solely pulmonary, but rather systemically enhanced in COPD and this pathological process might form a link between the pulmonary and cardiovascular manifestations of COPD [120]. This hypothesis is supported by the finding that emphysema severity, arterial stiffness, and skin elastin degradation in COPD patients significantly intercorrelate [120,121]. We therefore speculate that vitamin K might be a potential link between COPD and cardiovascular disease by protecting against systemic elastin degradation through an MGP-dependent pathway [122].

In addition to the prevention of macrovascular pathology, the literature is also suggestive of a protective effect of vitamin K against microvascular disease including deterioration of GFR and the development of microalbuminuria. Microalbuminuria is not only reflective of a disordered glomerular permeability [123], but is also a marker of systemic endothelial dysfunction including injury of the pulmonary microvasculature [124,125]. Microalbuminuria was associated with lung function decline, COPD-related hospital admissions, and COPD-related mortality independently from risk factors such as smoking [124]. Vitamin K2 has been demonstrated to increase nitric oxide (NO)-production thereby improving endothelial function [126]. In line, a human observational study demonstrated an inverse correlation between dp-ucMGP and FMD, a functional parameter of endothelial function [127], although this has not been confirmed in vitamin K supplementation trials [123,126].

Although most studies focused on the effects of vitamin K through pathways involving VKDPs, recently, increased attention has been paid to VKDP-independent effects of vitamin K. Vitamin K was demonstrated to have directly inhibiting effects on ferroptosis [128]. Ferroptosis is a pathological process of cell death due to iron-dependent lipid peroxidation [128]. Importantly, ferroptosis can be induced by cigarette smoke and is potentially involved in COPD pathogenesis [129,130]. Moreover, various studies demonstrated that vitamin K is a powerful anti-oxidant [119,131]. Nicotine induces intracellular ROS production and pre-treatment of VSMCs with vitamin K inhibited this process [131]. Vitamin K may also exert anti-inflammatory functions based on its inhibiting effects on proinflammatory cytokines in vitro and in vivo [132,133]. Reduced vitamin K intake and low plasma vitamin K1 levels were both associated with increased markers of inflammation in human cross-sectional studies [133,134], although clinical trials have not confirmed this association [132,134,135]. Oxidative stress and chronic inflammation are contributing mechanisms to the pathogenesis of COPD, cardiovascular disorders, osteoporosis, CKD, and sarcopenia [4,5,90,91,136,137] and vitamin K could form a possible link by counteracting these pathological processes. Future studies including animal models are, however, warranted to explore this.

Vitamin K deficiencies have been found in populations of subjects with COPD, arterial stiffness, osteoporosis, and CKD. Limited studies aimed, however, to unravel the underlying mechanisms by which these patients develop vitamin K deficiency. In all probability, reduced vitamin K consumption plays a role according to the numerous studies relating low vitamin K intake to disorders such as emphysema, vascular calcification, and the risk of bone fractures [21,39,40,92,96]. It is likely, however, that additional mechanisms contribute. Medium and long-chain forms of vitamin K2 can be produced by anaerobic bacteria in the gut [138,139,140] and concentrations of vitamin K2 in feces correlate with the pattern and abundance of intestinal bacteria [138,141]. There are indications that gut microbiota composition is involved in COPD pathogenesis [142,143]. COPD patients more frequently use antibiotics than subjects without COPD and in animal models, antibiotic therapy reduced vitamin K2-producing bacteria in the gut thereby influencing enteric vitamin K2 synthesis [144,145]. Additionally, increased gut permeability has been found during COPD exacerbations, leading to inflammation, enterocyte injury [146], and, potentially, disturbed vitamin K absorption. In addition to impaired intestinal absorption, increased vitamin K expenditure might be another explanation for vitamin K deficiency in COPD and co-existing chronic disorders. MGP is locally upregulated at sites of vascular calcification [147,148], possibly in an attempt to counteract this pathological process. Consequentially, in subjects with extensive vascular calcifications larger amounts of VKDPs have to be carboxylated [12], thereby exhausting the vitamin K storage. We hypothesize that enhanced elastin degradation in COPD could induce a similar effect on MGP upregulation, contributing to vitamin K deficiency.

Vitamin K supplementation would be a promising and low-cost intervention in COPD, although it should be emphasized that vitamin K has to be considered as an add-on. Vitamin K has an excellent safety profile, provided that subjects are not on VKA [149]. To the best of our knowledge, no interactions with current standard COPD treatments have been reported. To explore the potential beneficial effects of vitamin K on pulmonary and systemic manifestations of COPD, vitamin K supplementation trials are warranted. Based on previous studies, a clinical trial should preferably have a duration of at least three years in order to detect changes in outcome parameters. Although observational studies suggest that vitamin K2 is more potent than vitamin K1 for improving cardiovascular health, in clinical trials both vitamins appeared to be effective. It is important that the dosage of vitamin K supplementation is sufficiently high, since the available vitamin K will first be used for the carboxylation of hepatic proteins involved in coagulation and only the remaining vitamin K will be directed to extrahepatic tissues for carboxylation of VKDPs such as MGP and osteocalcin. It may be considered to determine whether the absorption of vitamin K is sufficient in COPD patients by measuring plasma levels of vitamin K following supplementation. Clinical endpoints of interest would be lung function decline, emphysema progression quantified by CT, aortic valve calcification, CAC, PWV, kidney function, bone fractures, and physical performance. Furthermore, it is important to identify those COPD patients that are most at risk of vitamin K deficiency. Taking the literature together, we suggest that patients with the emphysematous phenotype, vascular calcifications, and osteoporosis, have the highest risk to be vitamin K deficient and would potentially benefit most from vitamin K supplementation.

Moreover, we expect a synergistic effect when vitamin K is combined with vitamin D supplementation. Vitamin D upregulates MGP expression, a process that has also been demonstrated in the lungs [113]. Given that MGP is produced in the undercarboxylated form, vitamin D supplementation will increase the demand for vitamin K. It is expected that the combination of vitamin D-induced upregulation of MGP followed by vitamin K-dependent carboxylation of these proteins would result in the highest absolute increase in active MGP. Importantly, vitamin D monotherapy may deplete compromised vitamin K status in COPD patients, and we therefore propose that vitamin D should be combined with vitamin K supplementation.

In conclusion, based on present studies vitamin K might play a role in COPD pathogenesis and may form a link between COPD and both cardiovascular and bone health. Moreover, vitamin K deficiency is related to CKD as well as sarcopenia; however, evidence for a causal relationship is less convincing. Although clinical trials demonstrate positive effects on vascular calcification, arterial stiffness, and the risk of bone fractures, there is currently insufficient evidence for advising vitamin K supplementation for the prevention and treatment of multimorbidity in COPD. Basic research to unravel the underlying mechanisms, as well as large and well-designed clinical trials in COPD patients, are recommended.

## Figures and Tables

**Figure 1 jcm-12-01261-f001:**
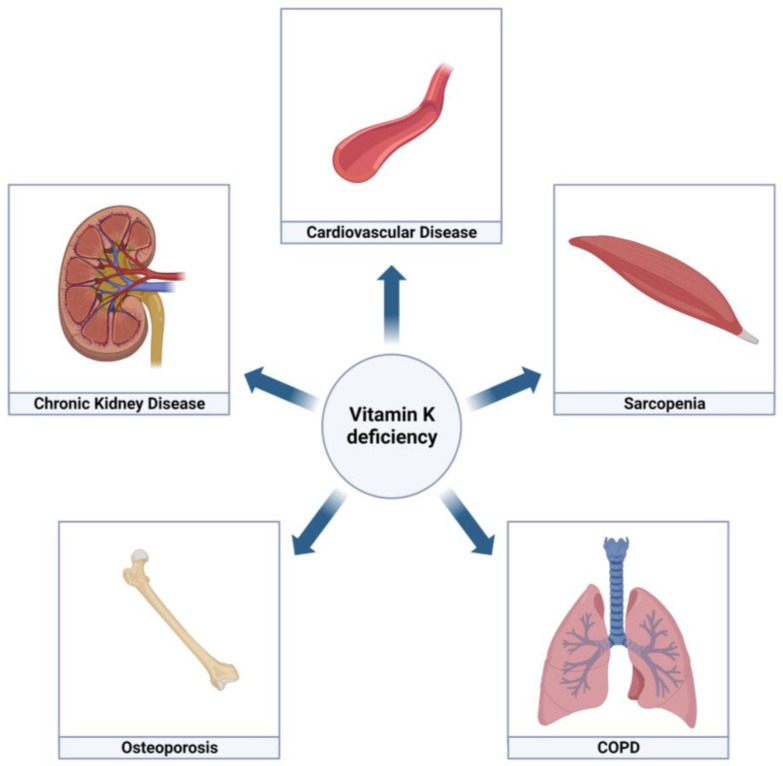
Vitamin K in multimorbidity of COPD. Created with BioRender.coms. accessed on 30 December 2022.

**Figure 2 jcm-12-01261-f002:**
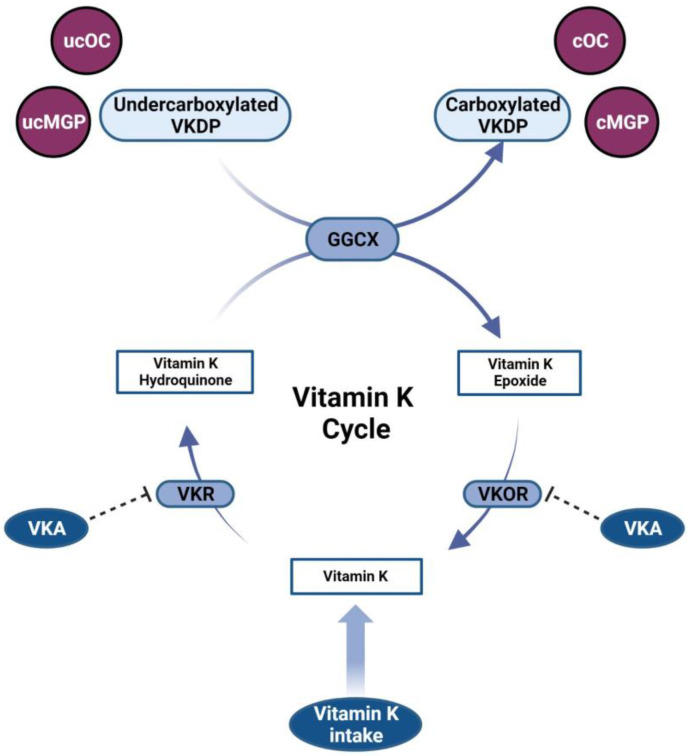
The vitamin K cycle. Vitamin K is reduced into the biologically active vitamin K hydroquinone. During the process of γ-carboxylation of vitamin K-dependent proteins (VKDPs), catalyzed by the enzyme γ-glutamyl carboxylase (GGCX), vitamin K hydroquinone is oxidized into vitamin K epoxide. Subsequently, in a two-step reduction catalyzed by vitamin K epoxide reductase (VKOR) and vitamin K reductase (VKR), vitamin K epoxide is converted back into vitamin K hydroquinone. Vitamin K antagonists (VKA) inhibit VKOR, thereby hampering the recycling of vitamin K. Carboxylated VKDPs including carboxylated osteocalcin (cOC) and carboxylated matrix Gla-protein (cMGP) exert biological activity. *ucMGP*: undercarboxylated matrix Gla-protein; *ucOC*: undercarboxylated osteocalcin. Created with BioRender.com. accessed on 28 January 2023.

**Figure 3 jcm-12-01261-f003:**
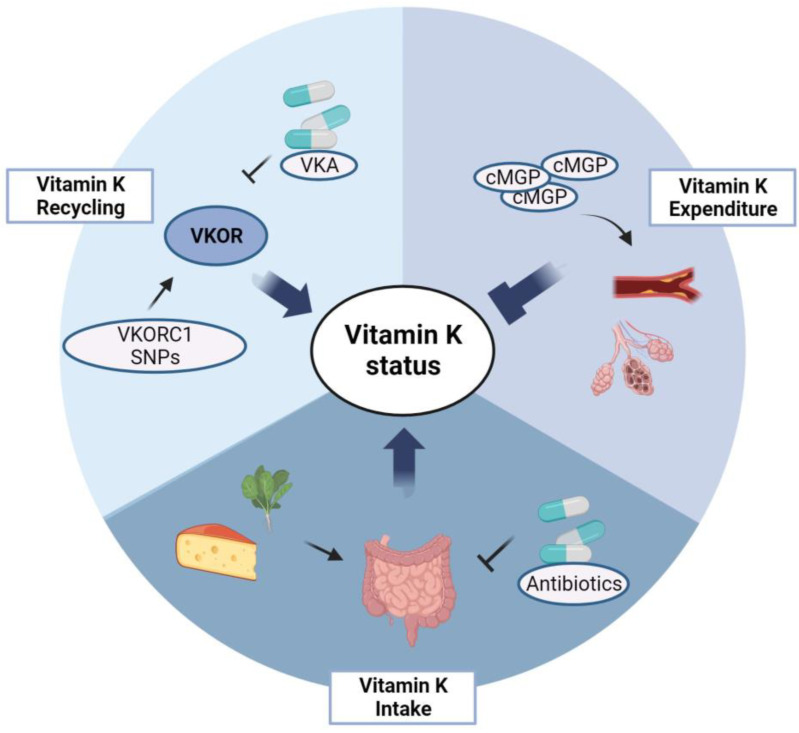
Determinants of vitamin K status. Vitamin K status is influenced by vitamin K intake, recycling, and expenditure. Vitamin K intake is determined by vitamin K consumption, intestinal absorption of vitamin K, and composition of gut microbiota, which can be influenced by use of antibiotics. Vitamin recycling is dependent on VKOR complex subunit 1 (VKORC1) haplotype and inhibition of vitamin K epoxide reductase (VKOR) by use of vitamin K antagonists (VKA). Increased vitamin K expenditure occurs in subjects with extensive vascular calcifications and, potentially, in subjects with pulmonary emphysema. In order to inhibit these pathological processes, the demand for vitamin K to activate VKDPs is increased, thereby exhausting the vitamin K storage. *cMGP*: carboxylated matrix Gla-protein; *SNPs*: single nucleotide polymorphisms. Created with BioRender.com. accessed on 30 December 2022.

## Data Availability

No new data were created or analyzed in this study. Data sharing is not applicable to this article.

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
