# Peer review of "The Pleiotropic Role of Vitamin K in Multimorbidity of Chronic Obstructive Pulmonary Disease"

_jcm, 2023, doi:10.3390/jcm12041261_

Round 1

Reviewer 1 Report

This manuscript is an excellent review about the potential role of vitamin K in the multimorbidity of COPD. The authors link COPD to its prevalent co-occurring chronic diseases with vitamin K as a connecting factor. Especially, the vitamin K-dependent protein MGP, which functions as a calcification inhibitor, was found to be related to most of the conditions and might account for the connection effects. Overall, I am sure this review will benefit the readers for future clinical studies.

Questions:

In Figure 2, changed the second VKOR to VKR: VKOR is not the only vitamin K reductase responsible for reducing vitamin K to vitamin K hydroquinone. FSP1, the new vitamin K reductase, has been identified recently. All these enzymes are commonly abbreviated as VKR.

Author Response

We would like to thank the reviewer for the comments. We agree with the suggestion and have amended figure 2 as well as the figure legend accordingly (line 112).

Reviewer 2 Report

The manuscript reviews the current litrature on possible role of vitamin K in COPD and some of the comorbidities (CVD, CKD, osteoporosis and sarcopenia). It is generally well written with current references. My only major concern is that the role of vitamin K is explained in COPD and comorbidities, but data about the role in COPD patient with comobridity is not clearly described or explained, so either data about it should be added if such exists, or if parts of disscussion imply it it should be reworded and written more clearly.

Author Response

We would like to thank the reviewer for the comments and suggestions. Data on the role of vitamin K in COPD patients are limited. To date, no studies have been performed on the effects of vitamin K on comorbidities in COPD patients per se. However, we think that the literature we have reviewed in this manuscript provides a clear rationale for future studies to investigate the effects of vitamin K on comorbidities in the COPD population. We have amended a sentence in the discussion that could be erroneously interpreted as if these studies have been performed yet (line 470). Also, we added a remark on this limitation in the first paragraph of the discussion (lines 440 to 444).

Reviewer 3 Report

Thank you for the opportunity to review this paper. It is well-written and comprehensive in it's review of the available literature and the common co-morbidities associated with COPD. No suggestions for revisions at this time. 

Author Response

We would like to thank the reviewer for the comments.